# The Stability of Hydroxyapatite/Poly-L-Lactide Fixation for Unilateral Angle Fracture of the Mandible Assessed Using a Finite Element Analysis Model

**DOI:** 10.3390/ma13010228

**Published:** 2020-01-04

**Authors:** Byungho Park, Bryan Taekyung Jung, Won Hyeon Kim, Jong-Ho Lee, Bongju Kim, Jee-Ho Lee

**Affiliations:** 1Department of Oral and Maxillofacial Surgery, College of Medicine, University of Ulsan, Asan Medical Center, Seoul 05505, Korea; bhp1111@nate.com; 2School of Dentistry, University of Detroit Mercy, Detroit, MI 48208, USA; tkjung1@gmail.com; 3Clinical Translational Research Center for Dental Science, Seoul National University Dental Hospital, Seoul 03080, Korea; wonhyun79@gmail.com (W.H.K.); leejongh@snu.ac.kr (J.-H.L.); 4Department of Oral and Maxillofacial Surgery, School of Dentistry, Seoul National University, Seoul 03080, Korea

**Keywords:** unilateral angle fracture, hydroxyapatite/poly-L-lactide, finite element analysis

## Abstract

Recently, a hydroxyapatite particle/poly-L-lactide (HA-PLLA) composite device was introduced as an alternative to previous fixation systems. In this study, we used finite element analysis to simulate peak von Mises stress (PVMS) and deformation of bone plates and screws with the following four materials—Ti, Mg alloy, PLLA, and HA-PLLA—at a unilateral mandibular fracture. A three-dimensional virtual mandibular model was constructed, and the fracture surface was designed to run from the left mandibular angle. Masticatory loading was applied on the right first molars. Stress was concentrated at the upper part and the neck of the screw. The largest PVMS was observed for Ti; that was followed by Mg alloy, HA-PLLA, and PLLA. The largest deformation was observed for PLLA; next was HA-PLLA, then Mg alloy, and finally Ti. We could rank relative superiority in terms of mechanical properties. The HA-PLLA screw and mini-plate deformed less than 0.15 mm until 300 N. Thus, we can expect good bone healing with usual masticatory loading six weeks postoperatively. HA-PLLA is more frequently indicated clinically than PLLA owing to less deformation. If the quality of HA-PLLA fixation is improved, it could be widely utilized in facial bone trauma or craniofacial surgery.

## 1. Introduction

It has been reported that the mandibular angle is the most vulnerable site for a mandibular fracture [1]. An angular fracture is associated with two segments (anterior and posterior segments). This fracture needs to be fixed at an accurate location in order to prevent non-union and mal-union in the long term. Thus, surgeons need to select an appropriate surgical technique and fixation material to stabilize the fractured mandible. With regard to technique, Champy’s concept is one of the most popular choices. In this approach, a four-hole mini-plate is fixed anteriorly and posteriorly along the external ridge until the fracture completely heals. Champy et al. demonstrated that the superior mandibular border was subjected to tension and splaying, whereas the inferior border was subjected to compression [2]. Therefore, two screws or more should be engaged along the external oblique ridge on each side of the fracture for stable fixation. In conjunction with an appropriate surgical technique, surgeons need to select the most reasonable material among the different options depending upon each case, because the prognosis might vary. Previous studies have suggested several different fixation materials to reduce the prevalence of postoperative complications. Among these materials, a titanium-based osteosynthesis system (Ti) has been recognized as the gold standard for facial bone surgery owing to its superior strength and stiffness [3]. However, some studies have demonstrated its disadvantages, such as palpability, migration, loosening, thermal irritability, infection, chronic pain, and the need for a secondary operation for removal [4,5]. To overcome the disadvantages of Ti and the magnesium (Mg) alloy, biodegradable polymers, such as poly-L-lactic acid (PLLA), have been suggested as alternative options. The Mg-alloy presents better strength than PLLA for internal fixation, but it is an unfavorable material for osteosynthesis because it produces hydrogen gas and damages the tissue [6,7]. On the other hand, biodegradable internal fixation products, such as PLLA, avoid the long-term foreign body reactions of metals and the disadvantages of Ti internal fixation [4]. PLLA has been successfully used in maxillofacial surgery [8]. However, there is controversy regarding biodegradable polymer application in maxillofacial areas because stability and low mobility have not yet been confirmed [9,10]. Tominaga et al. and Lee et al. support the clinical implementation of PLLA with evidence of both stability and low mobility, whereas Dorri et al. have questioned the results [11]. Although the PLLA fixation is not as strong as Ti or Mg alloys, its superiority in osteosynthesis is drawing the attention of surgeons. It will potentially be a good alternative option once a new variant with strength similar to that of Ti or Mg alloy is developed. Recently, a hydroxyapatite particle/poly-L-lactide (HA-PLLA; Osteotrans-MX; Takiron Co., Umeda, Japan) composite device was introduced as an alternative to previous systems. The strength of HA-PLLA is better than that of biodegradable PLLA because of the added hydroxyapatite, and it maintains the superiority of PLLA in osteosynthesis. Synthetic HA (Ca10(PO4)6(OH)2) has been widely used, owing to its osteoinductive characteristics [12], but it is not appropriate in areas with heavy loading owing to its brittleness and insufficient strength [13]. Osteotrans (Osteotrans-MX; Takiron Co., Umeda, Japan) is a bioresorbable and bioactive material that is made of forged unsintered hydroxyapatite (u-HA) and PLLA [14]. However, it is not as strong as Ti fixation materials. Therefore, further investigation is needed on the properties of HA-PLLA to determine if it is a safe and effective alternative. In this study, we evaluated the properties of HA-PLLA to determine if it is a reasonable substitute for Ti and Mg alloys and PLLA. In addition, we compared the stability of a HA-PLLA, four-hole mini-plate fixation to that of other materials for a unilateral mandibular fracture. Limited studies on HA-PLLA fixation are being performed at present; thus, assessment of HA-PLLA fixation for unilateral mandibular fracture involving the mandibular angle might increase the availability of this material for craniofacial surgery. Our team used a finite element analysis (FEA) program to simulate the peak von Mises stresses (PVMSs) and deformations of plates and screws at a unilateral mandibular fracture with the following four materials: Ti, Mg alloy, PLLA, and HA-PLLA.

## 2. Materials and Methods

### 2.1. Three-Dimensional Unilateral Mandibular Angle Fracture Model

A three-dimensional (3D) virtual mandibular model was constructed using Digital Imaging and Communications in Medicine (DICOM) data from cone beam computer tomography images (0.5 mm slices). The cortical and cancellous bones were considered isotropic, homogeneous, and linearly elastic according to Young’s modulus and Poisson’s ratio. For modeling and analysis, SolidWorks Software (Solidworks 2016, Dassault Systems SolidWorks Corp., Waltham, MA, USA) was used to construct the mini-plate and screw models, and the Hypermesh program (Altair Hyperworks v17.0, Altair Engineering Inc., Troy, MI, USA) was used to create a mesh of the bone and implant models. The cortical bone included 209,434 elements and 52,588 nodes, whereas the cancellous bone included 14,110 elements and 32,238 nodes. The screw included 40,525 elements and 8762 nodes, whereas the mini-plate included 42,800 elements and 8412 nodes. The model was simulated from node to node along the interconnecting elements throughout the mesh (Figure 1). An FEA program (ABAQUS CAE2016, Dassault Systems, Vélizy-Villacoublay, Yvelines, France) was used to assembly the bone and the implant parts. Using FEA, the following four types of implant systems were evaluated: (1) titanium screw and mini-plate (Optimus MF System^®^; Osteonic Ltd., Seoul, South Korea); (2) Mg alloy (Mg (92.4%), Ca (4.6%), and Zn (3.0%)); (3) PLLA polymer (INION CPS System; Inion Ltd., Tampere, Finland; copolymer of L-lactate (17%), D-lactate (78.5%), and TMC monomers (4.5%)); and (4) HA-PLLA (Osteotrans MX^®^; Teijin Medical Technologies Co., Ltd., Osaka, Japan; hydroxyapatite (40%) and PLLA (60%)). We used the uniform shapes of the mini-plates and screws for all materials to ensure consistency of the pattern. The material properties were the same for each of the plate and four screws. The mechanical properties of each material and bone are presented in Table 1 [15].

With regard to the study design, we set up experimental conditions that were consistent with those in previous studies [16,17]. The diameter and length of each screw were set to 2.0 and 6.0 mm, respectively, and the thickness of the plate was set to 2.0 mm. The cortical bone thickness was assumed to be 3.0 mm, and the fracture surface was designed to run from the mandibular angle to the external oblique ridge of the mandible. We set a simulated fracture on the mandibular angle starting from the gonion to the middle of the presumed third molar. To minimize the shearing effect of stress from bony contact, a 0.5 mm gap was maintained between the proximal and distal segments of the mandible. The locations of the screws and mini-plates were determined according to Champy’s concept. The mini-plates were placed at the load-bearing area of the external oblique ridge (Figure 1).

### 2.2. Evaluation of Virtual Masticatory Loading

Initial masticatory loading of 132 N was applied on the right first molars of the mandible under each condition (Figure 1), and two mandibular condyles were fixed to the glenoid fossa while masticatory loading was applied, consistent with previous studies [18,19]. Maximum stress distributions were calculated for the screws and mini-plates, and for the cortical and cancellous bones. From 200 N, the deformation under each condition was evaluated for every increment of 100 N in masticatory loading, until 1000 N was reached.

## 3. Results

### 3.1. Stress Distribution for the Titanium Plate and Screw System

The maximum PVMS values with the Ti system for the cortical and cancellous bones were 105.41 and 4.85 MPa, respectively (Table 2), and majority of the PVMS from masticatory forces was concentrated at the mandibular condylar neck on the non-fractured side (Figure 2a). With regard to the PVMS of the fixation materials, the screw had 214.71 MPa and mini-plate had 414.48 MPa (Table 2), and majority of the PVMS was concentrated at the upper central part of the mini-plate and the neck of the screw located closest to the fracture in the posterior mandibular segment (Figure 2b,c).

### 3.2. Stress Distribution for the Mg Alloy Plate and Screw System

The maximum PVMS values with the Mg alloy system for the cortical and cancellous bones were 106.89 and 4.93 MPa, respectively. These results were marginally different from those of the Ti system (Table 2). The majority of the PVMS was concentrated at the mandibular condylar neck on the non-fractured side, and it was similar to the finding with the Ti system (Figure 2d). With regard to the PVMSs of the fixation materials, the screw had 158.27 MPa and mini-plate had 229.53 MPa (Table 2). The stress concentration phenomena were similar to those of the Ti system (Figure 2e,f).

### 3.3. Stress Distribution for the PLLA Plate and Screw System

The maximum PVMS values in the PLLA system for the cortical and cancellous bones were 111.11 and 5.00 MPa, respectively (Table 2). Stress was concentrated at the mandibular condylar neck on the non-fractured side, similarly to the findings from the Ti and Mg alloy systems. However, the intensity was higher, as indicated by the red color code apparent in the data (Figure 2g). With regard to the PVMSs of the fixation materials, the screw had 83.62 MPa and mini-plate had 62.98 MPa (Table 2). The PVMS on the mini-plate was distributed between the superior posterior first holes, and it extended to the third hole. The majority of the PVMS was concentrated at the superior posterior second hole, but it was not as prominent as that with other materials (Figure 2h,i).

### 3.4. Stress Distribution for the HA-PLLA Plate and Screw System

The maximum PVMS values with the HA-PLLA system for cortical and cancellous bones were 109.08 and 4.99 MPa, respectively, not much different from the values for the other materials (Table 2). The findings of the conspicuous stress concentration at the condylar neck on the non-fractured side and its pattern were similar to the findings for the PLLA system (Figure 2j). With regard to the PVMSs of the fixation materials, the screw had 103.10 MPa and mini-plate had 82.20 MPa (Table 2). The values were remarkably lower than those of Ti and Mg alloys, but were slightly higher than the results for PLLA. The stress distribution was similar to that for the Ti and Mg alloy systems, but the concentration at the central part of the plate was less obvious (Figure 2k). Unlike the findings for the mini-plates, the stress distributions for the screws were similar regardless of the materials used (Figure 2c,f,i,l).

### 3.5. Deformation of the Materials in Masticatory Loading

Deformation was measured with the three axes in the Cartesian coordinate system. The largest deformation was observed for PLLA. Next was HA-PLLA, then Mg alloy, and finally, Ti. Deformation was larger in the mini-plates than in the screws for all materials. None of the materials showed deformation exceeding 0.6 mm (Table 3).

Data from Table 3 is visualized in Figure 3. For both the screws and mini-plates, deformation showed a linearly increasing trend as masticatory loading increased.

## 4. Discussion

The Ti system is the gold standard fixation system for facial bone trauma, but it is often associated with thermal sensitivity, inflammatory reactions, psychologic consideration, and possible fixation removal surgery. These issues have caused surgeons to consider biodegradable materials, such as PLLA and HA-PLLA. Our study focused on the stability of unilaterally fractured mandibles with HA-PLLA fixation. To evaluate stability, we used FEA to construct the mini-plate and screw models and simulate stress and deformation of a unilateral mandibular fracture with four materials. We expect a similar outcome between FEA simulations and clinical implementation. However, the inherent limitations of the FEA model, such as linearly elastic mechanical properties, and the isotropic and homogenous hypotheses [20], might deviate the simulated model from actual physiology and anatomy. Therefore, the sizes of the screws and mini-plates were unified regardless of the materials in order to evaluate the materials under equivalent conditions.

The inclusion of the third molars complicates a FEA simulation model owing to numerous possible combinations involving impaction depth, angulation, and ramus relationship. Additionally, the presence of the third molars might increase the chance of a mandibular fracture [21,22]. Because of these complexities, it is reasonable to exclude the third molars from a FEA model in order to perform a clinically relevant analysis. Therefore, in this study, we simplified the design for Champy’s concept by placing one straight mini-plate at the external oblique ridge area. In addition, we kept the model simple by using mini-plate and screw designs that were consistent with traditional designs, in accordance with previous studies [16,17].

Tate et al. concluded that the maximum strength for the cortical bone was 130 N of force one week postoperatively for unilateral mandibular angle fracture surgery and 251 N six weeks postoperatively [23]. The masticatory force was significantly lower on the fractured side than on the non-fractured side. Additionally, Gerlach et al. obtained a similar test result after surgery. They noted a force of 72 N at the first molar near the fracture site one week after surgery. According to these analyses, we can note that the non-fractured side had a high masticatory force after surgery. In the FEA model, we increased the masticatory loading at the first molar on the non-fractured side unilaterally from 132 to 1000 N, using increments of 100 N. The upper value of this range far exceeded 300 N, which was the maximum masticatory loading at six weeks postoperatively reported by Gerlach et al. [24]. As our team aimed to measure the stress distributions and deformations of various fixations, we tested above the routine masticatory loading of heathy adults of 700 N reported in the study by Ferrario et al. [25]. The simulated model demonstrated that the maximum PVMS was located at the cortical bones in the mandibular condylar neck on the non-fractured side and the screw neck. For the cancellous bones, the greatest PVMS was 5.00 MPa at 132 N of masticatory loading, and the value was the highest for PLLA, followed by HA-PLLA, Mg alloy, and Ti. A material with high stress distribution tends to have less deformation (Figure 3). For example, Ti, which can resist the highest PVMS, had the least deformation among the four materials.

Our study presented the maximum stress of 6.31 MPa for the cancellous bone at 132 N masticatory loading, which is higher than the maximum tensile stress of 3.9 MPa in the study by Misch et al. [26]. This discrepancy might have been caused by (1) the simplified linear assumption of the biomechanical behavior of the mandible, (2) the lack of an organic component and trabecular structure of cancellous bone in the 3D model, and (3) the absence of a cushioning effect by condylar movement and relevant muscles.

The maximum PVMS values for both the Ti and Mg alloy mini-plates and screws were larger than the stresses on the cortical bones, whereas the PLLA fixation stress was much smaller than the PVMS on the cortical bone. Additionally, the PVMS on the HA-PLLA screw was smaller than the PVMS on the cortical bone, but the difference was marginal. The majority of the PVMS was concentrated at the mandibular condylar neck on the non-fractured side, and the largest PVMS was associated with PLLA; that was followed by HA-PLLA, then Mg alloy, and finally, Ti. We observed that Ti reduced the load on the bone the most, but concentrated the stress on the fixation components. Ti and Mg alloy mini-plates showed more stress than the screws, and the majority of the stress was concentrated at the upper central part of the plate and the neck of the screws. However, different trends were observed for HA-PLLA and PLLA. For PLLA, the stress distribution on the mini-plate evenly appeared around the second hole and to the proximal segment, but it was not seen at the central part. For HA-PLLA, the stress distribution was similar to that for Ti and Mg alloy, but the concentration at the central part of the plate was less obvious.

According to our FEA simulation results, Ti and Mg alloy retained stress at the fixation components rather than transferring the mechanical force to the bones. On the other hand, with regard to PLLA and HA-PLLA, stress was concentrated at the condylar neck on the non-fractured side rather than at the fixation components (Table 2 and Figure 2g,j). These findings indicate that fixation involving biodegradable materials transferred masticatory loading to the adjacent biological components, as opposed to the fixation system containing Ti and Mg alloy. In addition, the complications, including nonunion, malunion, delayed union, and screw loosening, could happen if masticatory forces are exerted without the correct bone fixation [27]. As these materials do not hold the mechanical force sufficiently, adjacent structures, such as the temporomandibular joint, could potentially be compromised. With regard to the two biodegradable materials, HA-PLLA was superior to PLLA in the stress distribution at the fixation parts. In other words, stress to the biological components is lower with HA-PLLA than with PLLA. PLLA is already widely utilized in facial bone surgery, with reliable clinical outcomes owing to its sufficient strength to overcome the need for additional support for the fixation of fractures [5,28]. In addition, HA-PLLA shows biodegradability and superior biocompatibility over Ti or Mg alloys [29,30]. Therefore, it could be a highly recommended material in facial bone trauma and craniofacial surgery.

For all materials, the stress tended to be concentrated at the central part on the mini-plate, and the screw in the posterior mandibular segment closest to the fracture had the most PVMS. The PVMS was higher for the mini-plates than for the screws with Ti and Mg alloy, and the PVMS difference was larger with Ti than with Mg alloy. On the other hand, different trends were observed for HA-PLLA and PLLA, with larger PVMS for screws than for mini-plates. In terms of deformation magnitude, deformation tends to be greater for mini-plates than for screws under controlled conditions. Deformation was the highest with PLLA. Next was HA-PLLA; then, Mg alloy; and finally, Ti (Figure 3). With regard to HA-PLLA, the PVMS was higher for the screw than for the mini-plate, and this finding indicates that there might be more complications with screws than with mini-plates, possibly caused by screw fracture or loosening. Therefore, manufacturers might need to modify the design of HA-PLLA screws in order to reinforce its strength.

In our FEA simulation, we assumed masticatory loading on the non-fractured side while measuring the stress distribution from 132 to 1000 N after surgery. Applying 132 N (assumed masticatory loading in the first postoperative week) at the first molar on the non-fractured side deformed the materials between 0.058 and 0.112 mm [16]. On the other hand, applying 300 N (assumed masticatory loading in the sixth postoperative week), at the same site deformed the materials between 0.086 and 0.202 mm [24]. When masticatory loading of 700 N was applied according to the data for healthy adults by Ferrario et al. [25], deformation was 0.131–0.409 mm. At 1000 N (an extreme masticatory loading assumption), clinically acceptable deformation ranged between 0.166 and 0.565 mm and was reported for all four materials. According to the study by Søballe, the bone displacement distance should be less than 0.15 mm for fractured bones to have a good prognosis [31]. The amounts of stress and deformation influence bone matrix formation greatly [32]. The Ti screw and mini-plate deformed less than 0.15 mm until 800 and 700 N, respectively, and 600 N was the cut-off for Mg alloy. HA-PLLA tolerated masticatory loading of 300 N for both screw and mini-plate before relevant deformations, whereas the PLLA screw had more than 0.15 mm deformation at 300 N. At 132 N of masticatory loading, PLLA maintained less than 0.15 mm deformation. In other words, we can expect good bone healing with Ti and Mg alloys, and HA-PLLA at 300 N (expected masticatory loading six weeks post operation). It usually takes 6–8 weeks to form primary callus bone. Therefore, HA-PLLA can maintain better spatial stability with regard to the bone gap during primary callus bone formation when compared with the findings for PLLA [33]. On the other hand, PLLA can only endure 132 N (expected masticatory loading at one week postoperatively, and it might result in non-union or mal-union of the bone without control approaches, such as intermaxillary fixation. According to our results, HA-PLLA is more frequently indicated clinically than PLLA owing to less expected deformation. The durability of the HA-PLLA system could be improved through modifications of the screw design, such as greater diameter when compared with that of Ti or Mg alloy, and modification of the mini-plate design, especially in the central area. As HA-PLLA systems can achieve stability until completion of primary callus formation, these systems could be widely utilized in facial bone trauma or craniofacial surgery with modifications of the screw and mini-plate designs.

However, biodegradable materials have different biomechanical stabilities and fixation abilities depending on the duration of postoperative treatment. Therefore, future pre-clinical studies and biomechanical tests with cadaveric bone are required for preclinical efficacy evaluation and fixation safety depending on the duration of treatment.

## Figures and Tables

**Figure 1 materials-13-00228-f001:**
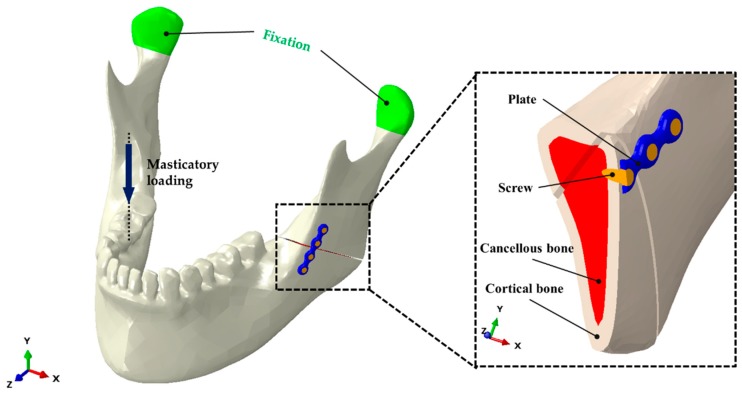
Unilateral angle fracture model of the mandible. Positioning of a four-hole plate and screws is shown at the left mandibular angle, and masticatory loading was applied to the right first molar (non-fractured side).

**Figure 2 materials-13-00228-f002:**
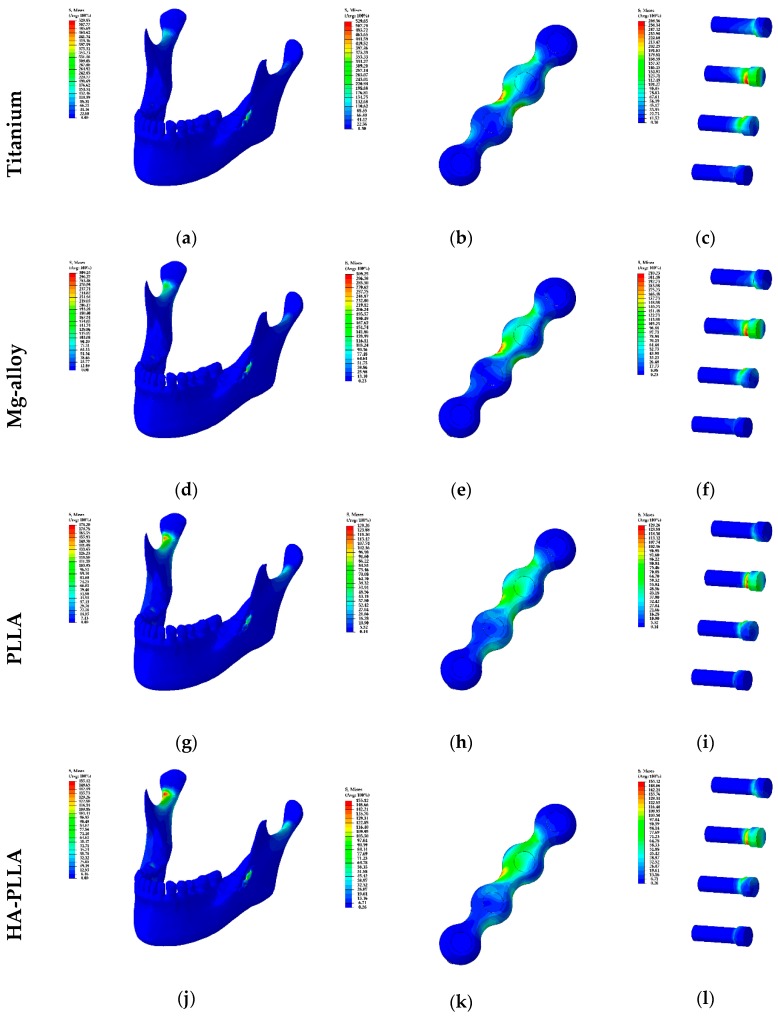
Visualization of Table 2: the distribution of stress in plate and screw systems based on materials is demonstrated from (**a**) to (**l**). The left column presents the stress on the mandible and the fixations while the middle column presents the stress of fixations: mini-plates and screws. The right column presents the stress on the screws solely.

**Figure 3 materials-13-00228-f003:**
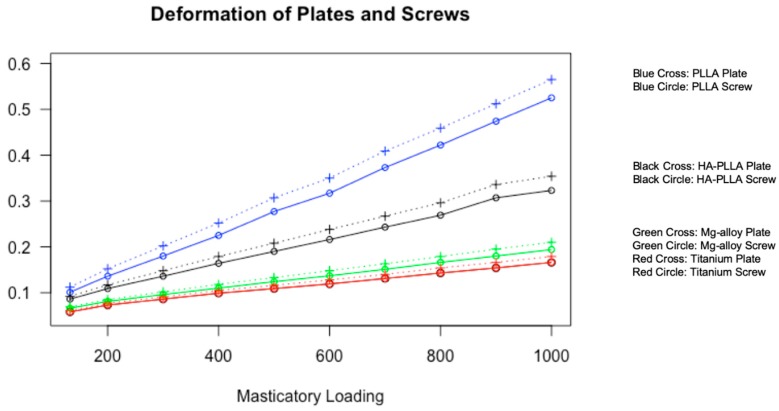
Visualization of material deformation against masticatory loading. All four materials show a linearly increasing trend of deformation for both screws and plates with different slopes. Titanium shows the least deformation, whereas PLLA shows the most deformation at the same pressure.

**Table 1 materials-13-00228-t001:** Mechanical properties of the materials and mandible for finite element analysis.

Components	Young’s Modulus (MPa)	Poisson’s Ratio
Titanium	96,000	0.36
Magnesium-alloy	45,000	0.29
PLLA (biodegradable) ^1^	3150	0.46
HA-PLLA ^2^	9701	0.317
Cortical bone	15,000	0.33
Cancellous bone	1500	0.3

^1^ Poly-L-lactic acid; ^2^ hydroxyapatite particle/poly-L-lactide.

**Table 2 materials-13-00228-t002:** The maximum peak von Mises stresses (PVMSs, MPa) and tensile stresses (MPa) for biological components and fixations according to the material at 132 N masticatory loading.

Components	Cortical Bone		Cancellous Bone		Screw	Plate
	PVMS	Tensile Stress	PVMS	Tensile Stress	PVMS	PVMS
Titanium	105.41	110.06	4.85	6.08	214.71	414.48
Mg-alloy	106.89	111.59	4.93	6.17	158.27	229.53
PLLA	111.11	141.92	5.00	6.31	83.62	62.98
HA-PLLA	109.08	114.01	4.99	6.26	103.10	82.20

**Table 3 materials-13-00228-t003:** Material deformation on masticatory loading according to the material: regardless of the material, deformation was greater in plates than in screws. The deformation level of HA-PLLA is between the two extreme levels of titanium and PLLA.

Masticatory Loading	Titanium (mm)		Mg-Alloy (mm)		PLLA (mm)		HA-PLLA (mm)	
	Screw	Plate	Screw	Plate	Screw	Plate	Screw	Plate
132 N	0.058	0.061	0.066	0.069	0.101	0.112	0.086	0.092
200 N	0.073	0.077	0.081	0.085	0.136	0.152	0.109	0.117
300 N	0.086	0.091	0.096	0.102	0.18	0.202	0.136	0.148
400 N	0.099	0.105	0.11	0.118	0.225	0.252	0.164	0.179
500 N	0.109	0.116	0.124	0.133	0.277	0.307	0.19	0.208
600 N	0.119	0.128	0.137	0.148	0.317	0.35	0.216	0.238
700 N	0.131	0.14	0.151	0.163	0.373	0.409	0.243	0.267
800 N	0.143	0.154	0.166	0.179	0.422	0.459	0.269	0.296
900 N	0.154	0.166	0.18	0.195	0.474	0.512	0.307	0.336
1000 N	0.166	0.179	0.194	0.21	0.525	0.565	0.323	0.354

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
