# Peer review of "The Stability of Hydroxyapatite/Poly-L-Lactide Fixation for Unilateral Angle Fracture of the Mandible Assessed Using a Finite Element Analysis Model"

_materials, 2020, doi:10.3390/ma13010228_

Round 1

Reviewer 1 Report

Dear authors,

  I analyzed you manuscript and I think that is of great interest for readers and that it falls within the scope of the journal.

Before publication I only recommend some corrections and integrations in the discussion section:

1 - In discussion you dealt about the possibility of mandibular fracture in case of presence of third molars, I suggest you to cite a recent literature review regarding the indication to third molar germectomy. In the same section you could add reference to other type of surgical interventions (Cavuoti, S. et al. Combined orthodontic-surgical management of a transmigrated mandibular canine: a case report. Angle Orthod. 2016, 86, 681-91) that are required in some typical populations (Perillo, L. Differences in craniofacial characteristics in Southern Italian children from Naples: a retrospective study by cephalometric analysis. Eur J Paediatr Dent. 2013, 14, 195-198).

2 - Line 237-239: You use this sentence: "These findings indicate that fixation involving biodegradable materials transferred masticatory loading to the adjacent biological components rather than the fixation system". If I got correctly the sense I suggest you to substitute the last two words with Ti and Mg fixation systems; otherwise please rewrite the sentence since it is not too clear. In the same paragraph I suggest you to add a sentence regarding the hypothetical complications that could happen during if masticatory forces are exerted without the correct bone fixation (for your convenience: Coviello V et al. Surgical ciliated cyst 12 years after Le Fort I maxillary advancement osteotomy: a case report and review of the literature Oral Surgery. 2017, 10, 165-170).

Best regards

Author Response

Dear reviewer

Comments and suggestions

I analyzed you manuscript and I think that is of great interest for readers and that it falls within the scope of the journal.

Before publication I only recommend some corrections and integrations in the discussion section:

In discussion you dealt about the possibility of mandibular fracture in case of presence of third molars, I suggest you to cite a recent literature review regarding the indication to third molar germectomy. In the same section you could add reference to other type of surgical interventions (Cavuoti, S. et al. Combined orthodontic-surgical management of a transmigrated mandibular canine: a case report. Angle Orthod. 2016, 86, 681-91) that are required in some typical populations (Perillo, L. Differences in craniofacial characteristics in Southern Italian children from Naples: a retrospective study by cephalometric analysis. Eur J Paediatr Dent. 2013, 14, 195-198).

Answer;

We have added more recent reference on your suggestion.

- row 193-194 : Additionally, the presence of the third molars might increase the chance of a mandibular fracture [21-22].

Line 237-239: You use this sentence: "These findings indicate that fixation involving biodegradable materials transferred masticatory loading to the adjacent biological components rather than the fixation system". If I got correctly the sense I suggest you to substitute the last two words with Ti and Mg fixation systems; otherwise please rewrite the sentence since it is not too clear.

Answer:

Thank you for pointing out the sentence in rows 239-241.

We have revised this sentence on your suggestion.

- row 239-241 : These findings indicate that fixation involving biodegradable materials transferred masticatory loading to the adjacent biological components rather than the fixation system containing Ti and Mg alloy

In the same paragraph I suggest you to add a sentence regarding the hypothetical complications that could happen during if masticatory forces are exerted without the correct bone fixation (for your convenience: Coviello V et al. Surgical ciliated cyst 12 years after Le Fort I maxillary advancement osteotomy: a case report and review of the literature Oral Surgery. 2017, 10, 165-170).

Answer:

We have added a new sentence and reference on your suggestion.

- row 241-243 : In addition, the complications including nonunion, malunion, delayed union and screw loosening could happen during if masticatory forces are exerted without the correct bone fixation [27].

Sincerely yours,

Reviewer 2 Report

Review for materials-688830-peer-review-v1

General Comments: The manuscript is nicely written and clear in its delivery. One major comment:

Please discuss the need for future studies using mechanical testing of these implants with cadaveric bone or using animal models at different healing durations. While there is great utility in FEA, it does not recapitulate the inherent biology, namely of the bone and soft tissue components of healing in animal models.

More specific comments and text editing/text corrections and listed below by line number.

More Specific Comments:

Title – None

Abstract

Line 22 – Indicate that the 4 materials are bone plates and screws.

Introduction

Line 43 – For “minimum of 2 screws”, specify if this means 2, 3, or 4 engaged cortices on either side of the fracture. Line 51 – Remove “metal” before “removal” Line 57 – Change “however” to “However” Line 72 – Remove “a” before “further”

Materials and Methods

Line 95 – Change “screw” to “implant” Line 95-99 – Are the screws and plates for each model made of the same material or are titanium screws used for all 4 different plates? Please specify.

Results – None

Discussion – See major comment above about the limitations of FEA.

Figures, Tables, and Legends – None

Author Response

Dear reviewer

Comments and suggestions

General Comments: The manuscript is nicely written and clear in its delivery.

One major comment:

Please discuss the need for future studies using mechanical testing of these implants with cadaveric bone or using animal models at different healing durations. While there is great utility in FEA, it does not recapitulate the inherent biology, namely of the bone and soft tissue components of healing in animal models.

Answer;

We have added new sentence on your suggestion.

- Line 290-293 : However, biodegradable materials have different biomechanical stability and fixation ability depending on the duration of postoperative treatment. Therefore, future pre-clinical studies and biomechanical tests with cadaveric bone are required for preclinical efficacy evaluation and fixation safety depending on the duration of treatment.

More specific comments and text editing/text corrections and listed below by line number.

More Specific Comments:

Abstract

Line 22 – Indicate that the 4 materials are bone plates and screws.

Answer;

We changed this sentence on your suggestion.

- Line 21-22 : In this study, we used finite element analysis to simulate Peak von Mises stress (PVMS) and deformation of bone plates and screws with the following four materials: Ti, Mg alloy, PLLA, and HA-PLLA at a unilateral mandibular fracture.

Introduction

Line 43 – For “minimum of 2 screws”, specify if this means 2, 3, or 4 engaged cortices on either side of the fracture.

Answer;

Two screws, at least, should be engaged at cortices on each side to confirm internal fixation of mandible fractures as based on Champy’s concepts. 

Then, we changed the sentence to clarify the meaning of “minimum of 2 screws” as follows:

- Line 44-45 : Therefore, two screws at least, or more should be engaged along the external oblique ridge on each side of the fracture for stable fixation.

Line 51 – Remove “metal” before “removal” Line 57 – Change “however” to “However” Line 72 – Remove “a” before “further”

Answer;

On your suggestion, We all edited Line 53, 58 and 73.

- Line 52 : remove “metal”

- Line 58 : Change “however” to “However”

- Line 73 : remove “a”

Materials and Methods

Line 95 – Change “screw” to “implant”

Answer;

As your opinion, we have changed “screw” to “implant”

- Line 96 : Using FEA, the following four types of implant systems were evaluated

Line 95-99 – Are the screws and plates for each model made of the same material or are titanium screws used for all 4 different plates? Please specify.

Answer;

We have revised this sentence on your suggestion.

- Line 102-103 : The material properties were the same for the plate and four screws for each of the four material.

Sincerely yours,

Round 2

Reviewer 1 Report

Dear authors,

  you satisfactorily addressed all my comments.

I suggest publication of the manuscript.

Best regards

Reviewer 2 Report

Review for materials-688830-peer-review-v2

The authors have nicely addressed all of my comments/edits.